# Breastfeeding Self-Efficacy in Postpartum Woman

**DOI:** 10.3390/healthcare13141690

**Published:** 2025-07-14

**Authors:** Maria Vitória da Silva, Rafaela Zumblick Machado, Valentina Fretta Zappelini Bittencourt, Maite Farias Bittencourt, Daniela Quedi Willig, Betine Pinto Moehlecke Iser

**Affiliations:** 1Medicine School, University of Southern Santa Catarina, Tubarão 88.704-000, Brazil; mariavsilvaa4@gmail.com (M.V.d.S.); rafazumblick@hotmail.com (R.Z.M.); valentinafretta@yahoo.com.br (V.F.Z.B.); maiteffb@hotmail.com (M.F.B.); dani.willig@hotmail.com (D.Q.W.); 2Graduate Program in Health Sciences, University of Southern Santa Catarina, Tubarão 88.704-000, Brazil

**Keywords:** breast-feeding, pregnant, self-efficacy, health promotion, postpartum woman

## Abstract

**Objective:** This study seeks to analyze the incidence of breastfeeding self-efficacy in postpartum woman who are undergoing prenatal care at the Family Health Strategy Units in the city of Tubarão, Santa Catarina (SC), Southern Brazil, from August to December 2022. **Methods:** An observational epidemiological study with a cross-sectional design was carried out with puerperal women—either primiparous or multiparous—who were followed in the municipal public network, along with their live-born children delivered at the maternity hospital of Hospital Nossa Senhora da Conceição in Tubarão, SC. Participants agreed to participate in the study. For data collection, an instrument developed by researchers and a validated instrument entitled Breastfeeding Self-Efficacy Scale-Short Form were used to assess the self-efficacy of breastfeeding. **Results:** The mean score of the Breastfeeding Self-Efficacy Scale (BSES) was 61.75 points (±6.39), indicating high self-efficacy. There was an association between the mean of the general score and/or the domains of the BSES with maternal characteristics. Women with higher education had greater technical mastery (*p* = 0.010), and those with previous breastfeeding experience and those who breastfed their children during the first hour of life had a higher average in the overall score and in the domains of the scale. In addition, those who planned the pregnancy (*p* = 0.024) and those who did not receive assistance from the milk bank (*p* = 0.047) had greater technical domain. **Conclusions:** In the present study, there was a predominance of high breastfeeding self-efficacy. It was verified that the personal and clinical aspects interfered in the self-efficacy of breastfeeding.

## 1. Introduction

Breastfeeding (BF) is of great importance for child growth and development, thus optimizing health as an adult [1]. The World Health Organization (WHO) recommends that children be exclusively breastfed for the first six months of life, and supplemented with other foods until the age of two [2]. Brazil is an excellent example of a society with high breastfeeding rates, with 45.7% of children under six months of age on exclusive breastfeeding and 53.1% of children aged 12 to 15 months on continued breastfeeding [3,4].

Breastfeeding is efficient in reducing infant mortality rates, as it is able to reduce mortality in children under five years of age by up to 13% [5]. Furthermore, it presents many protective factors throughout life, such as a reduction in obesity, high blood pressure, dyslipidemia, diabetes, and cancer [6]. Breastfed children have shown better cognitive performance and more developed social skills, which can be attributed to adequate nutrition and the emotional bond established during breastfeeding [7]. Above all, the bond between mother and infant is strengthened, which is fundamental for the child’s emotional development and the mother’s mental health [8].

The practice of breastfeeding is a complex process that goes beyond biological capability, and a woman’s decision to breastfeed or not also involves social, cultural, economic, and psychological factors [9]. Early interruption of breastfeeding may be related to a woman’s low confidence in her ability to breastfeed her baby, which is an important variable that influences the initiation and maintenance of breastfeeding [8,10].

Self-efficacy in breastfeeding, which refers to the mother’s belief in her ability to breastfeed, is a critical factor that can be modified through specific interventions. Therefore, in order to verify and understand the mother’s self-confidence and satisfaction in the breastfeeding process, it is useful to assess self-efficacy in breastfeeding, which can be modified using individual interventions for women [9]. Regarding health professionals, who are an essential factor in encouraging and supporting breastfeeding, the identification of mothers at risk of stopping breastfeeding can facilitate the development and evaluation of actions that promote breastfeeding [11].

As a result, the Breastfeeding Self-Efficacy Scale-Short Form (BSES-SF), an instrument used worldwide to assess maternal self-efficacy, corresponds to the shortened version of the Breastfeeding Self-Efficacy Scale (BSES) [12]. Using the BSES-SF, health professionals can identify women who are less likely to breastfeed and the areas in which they have a lower level of confidence and thus plan strategies to promote breastfeeding in a more individualized way [13].

From this perspective, given the countless factors that can interfere with breastfeeding, it is essential to identify these reasons that harm the practice of breastfeeding, with the goal of improving public policies for the protection and promotion of breastfeeding. Thus, this study aimed to evaluate breastfeeding self-efficacy in postpartum women who underwent prenatal care at Family Health Strategy Units (ESF) in the city of Tubarão, SC from August to December 2022.

## 2. Materials and Methods

This is an observational, cross-sectional study composed of postpartum women who underwent prenatal care at Family Health Strategy units in the city of Tubarão, Santa Catarina, Southern Brazil, between August and December 2022 and whose live births were hospitalized in the maternity ward of Hospital Nossa Senhora da Conceição (HNSC). This institution has achieved the title of Baby-Friendly Hospital since March 2001 and follows the precepts of the “Ten Steps to Encouraging Breastfeeding” [14]. The present study is part of an umbrella longitudinal study entitled “Intention, Self-efficacy in Breastfeeding, Duration of Breastfeeding, and Factors for Weaning: A Cohort Study”, which was approved by the Research Ethics Committee under opinion number 5,366,560.

Primiparous or multiparous postpartum women and their children born alive through normal birth or cesarean section with gestational age above 37 weeks were included. Participants agreed to participate in the study by signing the free and informed consent form and/or free and informed assent form. Postpartum women with the following conditions were excluded: multiple fetuses, changes in understanding or verbal expression that limited responses to data collection protocols, medical contraindication to breastfeeding due to infectious disease or medication use, foreigners, and mothers who destinated their children to adoption.

Data collection was carried out via electronic medical record analysis using the Philips Tasy^®^ system and in-person interviews during hospitalization in the rooming house. A protocol prepared by the researchers was used, comprising socioeconomic data (age, ethnicity, education, marital status, work activity, income), gynecological history (previous comorbidity, steady partner, use of contraceptive method), obstetric history (planned pregnancy, prenatal care—whether received and number of consultations, perceived family support, desire to breastfeed, previous breastfeeding, complications during pregnancy, delivery method, difficulty in breastfeeding, guidance on breastfeeding), and information on the newborn (time spent seeing the baby after birth, skin-to-skin contact, breastfeeding during the first hour of life, breastfeeding support, type of breastfeeding during hospitalization). The number of prenatal consultations followed the recommendations of the Brazilian Ministry of Health, which suggest a minimum of six visits to ensure adequate maternal and fetal monitoring [15].

The validated instrument entitled Breastfeeding Self-Efficacy Scale-Short Form—BSES-SF (Appendix A) [12] was used to assess maternal confidence in the ability to breastfeed. The BSES-SF is a Likert scale containing 14 questions divided into the following domains: Technical Domain (DT—eight items) and Intrapersonal Thinking Domain (DPI—six items). Each question presents five possible answers ranging from 1 to 5 points, including 1-totally disagree, 2-disagree, 3-sometimes agree, 4-agree, and 5-totally agree. The sum of the scores varies from 14 to 70 points. Breastfeeding self-efficacy is identified based on the sum of each question: low self-efficacy (14 to 32 points), medium self-efficacy (33 to 51 points), and high self-efficacy (52 to 70 points). The Cronbach’s alpha of the Brazilian version of the BSES-SF was 0.74, indicating high internal consistency, which was confirmed by the intraclass correlation coefficient, ranging from 0.69 to 0.78 [12]. The instrument was administered in Portuguese, within 24 to 48 h postpartum, considering the patient’s mode of delivery. The average time to collect data from each patient was 20 min.

The collected data were compiled in an electronic spreadsheet using Microsoft 365 Excel^®^ software and statistically treated using the Statistical Package for the Social Sciences (SPSS^®^) version 21.0 for Windows (IBM Corp. Armonk, NY, USA). Quantitative variables were described with measures of central tendency and dispersion, while qualitative variables were described in absolute numbers and proportions. To verify the association between the variables of interest, Pearson’s chi-square test or Fisher’s exact test was applied. The statistical analysis considered the assumptions underlying the applied tests. For the chi-square test, the independence of observations was ensured, and the expected frequencies in each cell were verified. However, in some contingency tables, expected cell counts were less than 5. In these cases, Fisher’s exact test was applied to ensure the statistical validity of the analysis. For the comparison of mean scores across instrument domains, the assumptions of normality (assessed using the Shapiro–Wilk test and graphical inspection) and homogeneity of variances (evaluated with Levene’s test) were verified to support the use of Student’s *t*-test or ANOVA, depending on the number of groups. In this case, the assumptions were met, and the application of non-parametric tests was not required. A significance level of 5% (*p* < 0.05) was adopted.

## 3. Results

Of the 120 women invited to participate, 95 met the inclusion criteria and were enrolled in the study. Among the 25 who were invited to participate in the study, 6 gave birth in the private healthcare system, 15 moved to another municipality, 1 placed the child for adoption, 1 was diagnosed with HIV (a formal contraindication to breastfeeding due to the risk of mother-to-child transmission), and 2 were admitted to the ICU. The mean maternal age was 27.88 years (Standard Deviation - SD ± 6.14 years), ranging from 16 to 44 years. Regarding sociodemographic characteristics, 73.6% (*n* = 70) self-identified as Caucasian. In total, 71.6% (68) had more than nine years of schooling, and only 4.2% (*n* = 4) were from the healthcare field. Most participants (80%, *n* = 76) had a partner, and 56.8% (*n* = 54) were employed. The average monthly family income was BRL 3162.92 (SD ± 1657.44)—corresponding to approximately USD 597 (SD ± 313), based on the average exchange rate at the time of the study.

In the sample evaluated, 83.7% (*n* = 77) of the participants did not present comorbidities prior to pregnancy. Of those who did (16.3%), the prevalent pathologies were asthma (31.1%), hypothyroidism (18.8%), and diabetes mellitus (6.3%). Regarding social habits, 6.3% (*n* = 6) used alcohol, and 4.2% (*n* = 4) were smokers.

Regarding gynecological characteristics, 89.9% (*n* = 85) reported the presence of a steady partner. Among women who used contraception (54.5%), the method of choice was oral contraceptives (80.4%). It was observed that most pregnant women did not plan their pregnancy (60%), but after the discovery, 89.5% (*n* = 85) of them received family support.

Regarding the previous breastfeeding of postpartum women, 64.2% (*n* = 61) of postpartum women had already breastfed during another pregnancy. Of these, 27.4% (*n* = 26) had some difficulty, mainly related to incorrect attachment (30.8%), low quantity of milk (23.1%), and type of nipple (15.7%). Regarding the desire to breastfeed during this pregnancy, 98.9% (*n* = 94) of the women said they had it. However, 60% (*n* = 57) of them reported not having received guidance on breastfeeding during prenatal care, and those who were advised (40%) received information from the general practitioner responsible for the ESF (50%) and the unit’s nurses (37.5%).

Regarding birth characteristics, the majority of women reported no complications during pregnancy (61.1%) or childbirth (95.8%). Almost the entire sample (97.3%) reported seeing their child for the first time in the delivery room, and 94.7% (*n* = 90) experienced skin-to-skin contact within the first hour of life.

Regarding postpartum breastfeeding, 75.8% (*n* = 72) of mothers initiated breastfeeding within the first hour, 94.7% (*n* = 90) felt supported to start breastfeeding, and 74.7% (*n* = 71) received breastfeeding guidance while in the hospital. The majority of newborns (94.7%) were on exclusive breastfeeding (EBF) during their hospital stay.

The analysis of the BSES-SF scale scores revealed a mean of 60.57 points (SD ± 6.09), with a range from 44 to 70 points. A high level of breastfeeding self-efficacy was observed in 93.7% (*n* = 89) of the sample, 6.3% (*n* = 6) exhibited a medium level, and no participants were classified as having a low level of self-efficacy. When comparing the classification of the self-efficacy scale and maternal characteristics, no statistical differences were found (Table 1). However, numerically, higher proportions of women over 27 years of age, with higher education, a partner, previous experience with breastfeeding experience, and a greater number of prenatal consultations were observed among those with high breastfeeding self-efficacy.

Table 2 presents the comparison of the means of the general score of the self-efficacy scale and the domains with the maternal characteristics. There was a statistical difference in all variables, except maternal age. Women with higher education had a higher average in the technical domain (*p* = 0.010). Women with previous experience with breastfeeding had a higher average score in the general score (*p* = 0.013), technical domains (*p* = 0.045), and intrapersonal thoughts (*p* = 0.019). Postpartum women who planned their pregnancy had a higher average in the technical domain (*p* = 0.024), while women who did not need the help of the milk bank had a higher average in the technical domain (*p* = 0.047). Postpartum women who breastfed during the first hour of life had a higher average in the general score (*p* = 0.005), the technical domain (*p* = 0.034), and intrapersonal thoughts (*p* = 0.005).

## 4. Discussion

It is appropriate to outline the profile of postpartum women because it is known that practices related to breastfeeding have been shown to be associated with several factors, including maternal age. The mean age found in the present study corroborates findings in the literature of 27.88 (SD ± 6.14) years [16]. The comparison of breastfeeding self-efficacy classifications among postpartum women revealed no statistically significant difference, suggesting that maternal age, as an isolated variable, does not influence the level of maternal breastfeeding self-efficacy. Similarly, a study conducted in Australia found no significant association between maternal age and breastfeeding self-efficacy levels [17]. However, studies published by Chaves et al. and França et al. reported that adolescent mothers (<20 years old) tend to breastfeed for shorter durations compared to adult mothers, possibly due to lower self-confidence leading to earlier weaning [18,19]. On the other hand, adult women may have greater experience and knowledge regarding breastfeeding, which could explain their higher level of self-efficacy [20].

Another important factor for breastfeeding is the marital stability of the parents, which is considered a positive influence on the breastfeeding process, as it promotes a calm and welcoming environment for the woman to be able to breastfeed with physical and emotional support given to the postpartum woman [21,22]. In the present study, although most women reported having a partner, no significant differences were observed in breastfeeding self-efficacy based on partner status.

In the present study, the majority of postpartum women interviewed underwent adequate prenatal care as recommended by the Ministry of Health, thus favoring greater self-efficacy. These consultations constitute a set of clinical and educational procedures that aim to monitor the health of the mother and child. Correct monitoring of pregnant women allows early detection and treatment of morbidities, reducing negative outcomes for the mother and child. Furthermore, correct monitoring during prenatal care promotes greater success in the breastfeeding process [23]. Therefore, the present study is consistent with greater breastfeeding self-efficacy in women who carried out ideal monitoring of prenatal consultations, which provided better guidance during the gestational period, including control of correct practice and factors that can influence the breastfeeding process. maternal. It is noteworthy, therefore, that the assistance provided during prenatal care has a beneficial influence on breastfeeding behavior and practices. Thus, there is no interruption due to lack of information, and these postpartum women feel welcomed in the process.

Postpartum women who had a vaginal birth had a higher self-efficacy score in this study, but the result lacked statistical significance. However, the opposite result was found in research in which cesarean section was associated with lower breastfeeding self-efficacy score given that, post-surgical birth, the mother feels pain and discomfort that makes it difficult to position the baby at the breast [24].

When evaluating breastfeeding self-efficacy levels, high self-efficacy was observed in 95.3% of the postpartum women interviewed. A similar result was reported in a study conducted in Northeast Brazil, which found high self-efficacy in 83.3% of the sample [25]. Furthermore, analysis of other data revealed that certain variables were associated self-efficacy, including women’s preparation during prenatal consultations, breastfeeding experience, partner support, and level of education.

When evaluating the overall self-efficacy score and its domains in relation to maternal and obstetric characteristics, all variables analyzed showed statistically differences in at some aspect, except for maternal age. These findings suggest an association between higher education and increased breastfeeding self-efficacy, consistent with previous reports in the literature [26]. Furthermore, these women showed greater self-efficacy in the technical domain. In this context, it was observed that the education level of postpartum women facilitates learning about breastfeeding during the educational strategies that are offered in the prenatal and postpartum period, also considering that more educated women have greater access to information about breastfeeding.

In the present study, postpartum women who had previously breastfed had higher overall scores in self-efficacy, the technical domain, and the intrapersonal thoughts domain. According to Bandura’s Theory, self-efficacy beliefs are fueled by lived experiences [27]. In addition to this, this finding is in line with that elucidated by Lopes et al., which states that previous breastfeeding experiences can influence the effectiveness shown by these women to breastfeed their current child [28]. Therefore, it is clear that this experience is relevant. In addition to being a process that requires correct practice, it is an adaptive process for both the mother and the child. In this regard, mothers who had previously gone through this experience adapted more easily.

Higher self-efficacy scores were also observed, similar to this study, in women who had a planned pregnancy [29,30]. In view of this, unwanted pregnancy may be a factor associated with reduced breastfeeding duration by exposing women to psychosocial stress, which may inhibit the practice of desirable health behaviors. On the other hand, women who want a pregnancy are, in a certain way, more curious to improve their knowledge and practices about the entire pregnancy process, including breastfeeding.

Early initiation of breastfeeding is defined as beginning the breastfeeding process within one hour after birth. This practice is associated with several health benefits, enhanced immune defense, reduced risk of diarrhea, and increased child survival rates [31]. In the present study, the majority of participants reported initiating breastfeeding during the first hour of the newborn’s life. These women also presented higher self-efficacy scores overall and in both domains. Although causality cannot be established, early contact may contribute to greater maternal satisfaction and bonding with the newborn by promoting feelings of connection, care, and affection.

Regarding the use of the milk bank, the postpartum women who needed this means had less technical proficiency. In this context, it is possible that these women experienced certain challenges related to breastfeeding. In view of this, it is important to remember that the milk bank is a way for postpartum women to also receive information about correct breastfeeding practices in order to contribute and/or solve the problems for which they needed this assistance.

The limitations of the present study include the relatively small sample size from a single center. Some participants were lost due to refusal to participate, changes of residence, or deliveries occurring in private healthcare facilities. Additionally, the cross-sectional design limits the ability to establish causal relationships. While these aspects may constrain the generalizability of the findings, the data still offer valuable insights into breastfeeding self-efficacy among postpartum women. These results can serve as a foundation for future studies with larger, more diverse samples and longitudinal designs, which would allow for better generalization and a deeper understanding of causal pathways.

## 5. Conclusions

Based on the BSEF-SF assessment of maternal and obstetric characteristics, factors such as education level, previous breastfeeding experience, pregnancy planning, use of a human milk bank, and breastfeeding within the first hour of life were associated with higher breastfeeding self-efficacy. Furthermore, the present study showed a predominance of high self-efficacy levels, indicating a positive outlook for breastfeeding practices in this population.

The use of the BSES-SF can enable health professionals to implement more targeted interventions aimed at promoting and supporting exclusive breastfeeding, particularly among women without prior breastfeeding experience. In clinical practice, the scale serves as a valuable tool for identifying the health needs of women during the prenatal and postpartum periods. Its application can contribute to the development of strategies to enhance maternal self-efficacy and support effective breastfeeding practices.

It is therefore suggested that health teams work to strengthen connections with families and community support networks, helping to better prepare them to support breastfeeding women. Additionally, this study may inform future intervention research and guide comprehensive professional practice in breastfeeding support, ultimately aiming to improve the quality of care provided. Future research should focus on longitudinal studies with larger and more diverse populations to explore the long-term effects of self-efficacy interventions and their impact on breastfeeding outcomes.

## Figures and Tables

**Table 1 healthcare-13-01690-t001:** Classification of the breastfeeding self-efficacy scale related to the characteristics of postpartum women treated in the maternity ward of a hospital in the south of Santa Catarina from August to December 2022. *N* = 95.

	Sample		Self-Efficacy		
	Frequency *n* (%)	High *n* (%)	Moderate *n* (%)	Total *n* (%)	*p*-Value *
**Age group (years)**					0.700
<27	41 (43.2)	39 (95.1)	2 (4.9)	41 (100)	
≥27	54 (56.8)	50 (92.6)	4 (7.4)	54 (100)	
**Ethnicity**					0.335
Caucasian	70 (73.6)	64 (91.4)	6 (8.6)	70 (100)	
Non-Caucasian	25 (26.3)	25 (100)	-	25 (100)	
**Education (years studied)**					0.671
≥9	68 (71.6)	63 (92.6)	5 (7.4)	86 (100)	
<9	27 (28.4)	26 (96.3)	1 (3.7)	9 (100)	
**Marital status**					1.000
With a companion	76 (80.0)	71 (93.4)	5 (6.6)	76 (100)	
Without a companion	19 (20.0)	18 (94.7)	1 (5.3)	19 (100)	
**Previous breastfeeding**					0.183
Yes	61 (64.2)	59 (96.7)	2 (3.3)	61 (100)	
No	34 (34.8)	30 (88.2)	4 (11.8)	34 (100)	
**Desired pregnancy**					0.680
Yes	38 (40.0)	35 (92.1)	3 (7.9)	38 (100)	
No	57 (60.0)	54 (94.7)	3 (5.3)	57 (100)	
**Number of prenatal consultations**					1.000
≥6	75 (78.9)	70 (93.3)	5 (6.7)	75 (100)	
<6	20 (21.1)	19 (95)	1 (5.0)	20 (100)	
**Delivery method**					1.000
Vaginal	51 (53.7)	48 (94.1)	3 (5.9)	51 (100)	
Cesarean section	44 (46.3)	41 (93.2)	3 (6.8)	44 (100)	

* *p*-Value from Fisher’s exact test.

**Table 2 healthcare-13-01690-t002:** Comparison of the means of the general score of the breastfeeding self-efficacy scale and its domains with the characteristics of postpartum women treated in the maternity ward of a hospital in the south of Santa Catarina from August to December 2022. *N* = 95.

Characteristics of the Sample		GS			TD			DPI	
	Mean	SD ^#^	*p*-Value	Mean	SD	*p*-Value	Mean	SD	*p*-Value
Age (years)									
≥27	61.26	5.96	0.206	35.72	3.78	0.268	25.53	2.93	0.307
<27	59.66	6.21		34.87	3.53		24.87	3.30	
Education (years)									
≥9	62.30	5.29	0.082	36.89	3.01	0.010 *	25.40	3.34	0.761
<9	59.88	6.28		34.75	3.73		35.29	3.02	
Previous breastfeeding									
Yes	61.72	5.48	0.013 *	35.91	3.22	0.045 *	25.80	2.87	0.019 *
No	58.50	6.64		34.35	4.21		24.26	3.27	
Planned pregnancy									
Yes	61.95	6.00	0.072	36.39	3.51	0.024 *	25.55	3.35	0.444
No	59.65	6.03		34.66	3.62		25.05	2.92	
Use of a milk bank									
Yes	58.18	6.82	0.074	33.76	4.33	0.047 *	24.41	3.85	0.219
No	61.09	5.84		35.70	3.43		25.43	2.90	
Breastfed during the first hour of life									
Yes	61.56	5.54	0.005 *	35.80	3.26	0.034 *	25.75	2.85	0.005 *
No	57.48	6.80		33.95	4.50		23.69	3.36	

SD ^#^: standard deviation; * *p*-value < 0.05; GS: general score; TD: technical domain; DPI: Intrapersonal Thinking Domain.

## Data Availability

Ethical, legal, and privacy issues are present. The study is in accordance with the consent provided by participants on the use of confidential data. The publication of such data does not compromise the anonymity of the participants or breach local data protection laws.

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
