# Peer review of "Breastfeeding Self-Efficacy in Postpartum Woman"

_healthcare, 2025, doi:10.3390/healthcare13141690_

Round 1
Reviewer 1 Report
Comments and Suggestions for Authors
Journal: Healthcare (ISSN 2227-9032)
Manuscript ID healthcare- 3691133
Type: Article
Breastfeeding self-efficacy in postpartum woman
Authors: Maria Vitória da Silva, Rafaela Zumblick Machado, Valentina Fretta Zappelini Bittencourt, Maite Farias Bittencourt, Daniela Quedi Willig, Betine Pinto Moehlecke Iser *
Please consider providing some more specific comments addressing the
following points:
- What is the main question addressed by the research?
The main question addressed by the study was to identify the personal and clinical determinants that impact maternal confidence in breastfeeding during the immediate postpartum period among women in Southern Brazil.
- Do you consider the topic original or relevant to the field? Does it address a specific gap in the field? Please also explain why this is/ is not the case. What does it add to the subject area compared with other published material?
The topic of the article is very interesting and relevant to the field of maternal and child health. Breastfeeding self-efficacy is a well-established predictor of breastfeeding initiation and duration; however, this study addresses a specific gap by focusing on a Brazilian population of postpartum women within the context of public health units and by examining a comprehensive set of influencing factors—such as education, previous experience, pregnancy planning, use of milk banks, and timing of breastfeeding initiation. The results obtained from this research could help tailor interventions to specific communities and healthcare settings, which is essential for improving breastfeeding outcomes in the target population. While the work is generally well executed, there are several areas that require improvement. Please find my comments and suggestions below:
INTRODUCTION
- Please clarify in greater detail the specific contribution and originality of this research, and explain how it differs from previously conducted studies.
- Although the introduction is relatively well written, it would be beneficial to expand it further and discuss in more detail all the benefits of breastfeeding for both women and children, including psychological aspects.
- Please remove the bold formatting from lines 40–42.
MATERIALS AND METHODS
• What specific improvements should the authors consider regarding the
methodology?
- Please provide a more detailed explanation of the questions included in the BSES-SF (the version used in this study), or include a copy of the questionnaire as a supplementary appendix to the article.
- It would be helpful to specify the language in which the questionnaire was administered—whether it was in English or Portuguese.
- Regarding specific methodological improvements, it would be beneficial to increase the sample size (an appropriate sample size could be calculated prior to conducting the study using statistical tests such as G*Power), and to consider alternative study designs, such as a longitudinal rather than a cross-sectional design.
- Have any other variables related to breastfeeding self-efficacy in postpartum women been examined? It might be beneficial to include additional variables in the study, such as social support, cultural background, or the presence or absence of postpartum depression or anxiety.
RESULTS
- The authors should include a brief statement regarding the response rate (i.e., the percentage of invited participants who agreed to participate), to enable readers to assess the representativeness of the sample.
- In addition, the authors should report the percentage of patients who were excluded from the study due to changes of address.
- With respect to educational level (years of study), is there any information regarding the type of education (medical vs. non-medical)? It would be of interest to determine whether there are differences between participants with medical and non-medical backgrounds in terms of their knowledge, confidence in breastfeeding ability, and breastfeeding self-efficacy.
DISCUSSION
- The discussion section should be strengthened and expanded with more appropriate and up-to-date references.
- The limitations and shortcomings of this study should be discussed in more details, such as the small sample size and the cross-sectional design.
CONCLUSION
• Are the conclusions consistent with the evidence and arguments presented
and do they address the main question posed? Please also explain why this
is/is not the case.
- There is no need to repeat the main findings and results of the study (lines 283–287).
- It would be beneficial to emphasise the value and contribution of this study to the field, as well as to provide recommendations for future research.
- The conclusions are consistent with the evidence and arguments presented because the study finds a predominance of high breastfeeding self-efficacy among participants and demonstrates that factors such as higher education, previous breastfeeding experience, planned pregnancy, absence of need for milk bank support, and early initiation of breastfeeding are associated with higher self-efficacy scores (BSES-SF assessment). These findings directly address the main research question and are supported by statistical analysis (Table 1. and Table 2.).
REFERENCES
• Are the references appropriate?
- The reference formatting is not consistent. I recommend reviewing the references to ensure compliance with the journal’s author guidelines and excluding all references older than 10 years unless they are essential (references under the numbers 7,11,15,17,18,19,20,29,31).
- Additionally, it would be beneficial to include more relevant and recent scientific references, preferably published within the last five years, especially in the discussion section.
- Any additional comments on the tables and figures.
- There are no figures included in the manuscript; therefore, providing some visual representations would be beneficial to enhance reader comprehension.
- There are two tables included in the manuscript, which are well-organized and comprehensive.
The manuscript would benefit from further language refinement. Utilizing a professional English-language editing service is recommended.
Author Response
Review 1
INTRODUCTION
- Please clarify in greater detail the specific contribution and originality of this research, and explain how it differs from previously conducted studies.
- Although the introduction is relatively well written, it would be beneficial to expand it further and discuss in more detail all the benefits of breastfeeding for both women and children, including psychological aspects.
- Please remove the bold formatting from lines 40–42.
Response: We appreciate the Editor’s comment. We revised and improved the introduction section and added some parts and references according to the reviewers' comments and suggestions. All modifications are in red.
MATERIALS AND METHODS
- Please provide a more detailed explanation of the questions included in the BSES-SF (the version used in this study), or include a copy of the questionnaire as a supplementary appendix to the article.
Response: Thank you for pointing this out and allowing us to elaborate further. We have included additional details regarding the instrument, and a copy of the questionnaire has been added to the manuscript as Appendix A.
It would be helpful to specify the language in which the questionnaire was administered—whether it was in English or Portuguese.
Response: Thanks for your comment. The questionnaire's language was incorporated (Line 114)
- Regarding specific methodological improvements, it would be beneficial to increase the sample size (an appropriate sample size could be calculated prior to conducting the study using statistical tests such as G*Power), and to consider alternative study designs, such as a longitudinal rather than a cross-sectional design.
Response: We thank the reviewer for this valuable observation and the opportunity to clarify. We fully agree that increasing the sample size and adopting a longitudinal design would enhance the robustness of the findings. However, as this study has already been conducted, it is not feasible to implement such modifications at this stage. It is important to note that the present research is a secondary analysis derived from a prospective cohort study that followed pregnant women from the third trimester through the first year of the child’s life. For this specific analysis, breastfeeding self-efficacy was assessed in the immediate postpartum period (within 24–48 hours), during hospitalization after delivery. The sample included all women from the cohort who received prenatal care in primary healthcare units and gave birth in the designated reference maternity hospital in the region. Therefore, expanding the sample retrospectively is not possible. Nevertheless, in response to the reviewer’s suggestion, we have explicitly acknowledged the limitation related to sample size in the revised manuscript (see Limitations section).
- Have any other variables related to breastfeeding self-efficacy in postpartum women been examined? It might be beneficial to include additional variables in the study, such as social support, cultural background, or the presence or absence of postpartum depression or anxiety.
Response: We thank the reviewer for this thoughtful comment. As mentioned earlier, the present study is derived from a larger prospective cohort project. While some additional variables potentially related to breastfeeding self-efficacy were collected as part of the broader study, they were not included in the current analysis, which was focused on the immediate postpartum period and a specific set of research questions. These additional aspects will be explored in future stages of the project. We have added a brief clarification in the manuscript to reflect this context (see page 2, lines 81-84).
RESULTS
- The authors should include a brief statement regarding the response rate (i.e., the percentage of invited participants who agreed to participate), to enable readers to assess the representativeness of the sample.
- In addition, the authors should report the percentage of patients who were excluded from the study due to changes of address.
- With respect to educational level (years of study), is there any information regarding the type of education (medical vs. non-medical)? It would be of interest to determine whether there are differences between participants with medical and non-medical backgrounds in terms of their knowledge, confidence in breastfeeding ability, and breastfeeding self-efficacy.
Response: We thank the reviewer for the opportunity to clarify these important aspects. The requested information has been added at the beginning of the Results section, on page 3 (lines 136–140), to enhance the clarity and completeness of the findings.
DISCUSSION
- The discussion section should be strengthened and expanded with more appropriate and up-to-date references.
- The limitations and shortcomings of this study should be discussed in more details, such as the small sample size and the cross-sectional design.
Response: Thank you for these valuable comments. We have revised and expanded the Discussion section to provide a more robust interpretation of the findings. Additionally, we have elaborated further on the study’s limitations, specifically addressing the small sample size and the constraints associated with the cross-sectional design (see Discussion section, page 8, lines 291-299).
CONCLUSION
- There is no need to repeat the main findings and results of the study (lines 283–287).
- It would be beneficial to emphasise the value and contribution of this study to the field, as well as to provide recommendations for future research.
Response: We thank the reviewer for these helpful suggestions. In response, we have revised the Conclusion section to avoid repetition of the main findings, in accordance with the recommendation. We also added a brief statement highlighting the study’s contribution to the field and included suggestions for future research directions, aiming to enhance the relevance and applicability of our findings (see page 9, lines 308-320).
REFERENCES
- The reference formatting is not consistent. I recommend reviewing the references to ensure compliance with the journal’s author guidelines and excluding all references older than 10 years unless they are essential (references under the numbers 7,11,15,17,18,19,20,29,31).
- Additionally, it would be beneficial to include more relevant and recent scientific references, preferably published within the last five years, especially in the discussion section.
Response: We appreciate the reviewer’s careful evaluation of the references. We have thoroughly reviewed the reference list to ensure full compliance with the journal’s author guidelines. Regarding the references older than 10 years, we chose to retain them because they represent seminal Brazilian and international studies that are still highly relevant to the subject and widely cited in current literature.
Reviewer 2 Report
Comments and Suggestions for Authors
Manuscript: Breastfeeding self-efficacy in postpartum woman
Summary
This cross-sectional study examined breastfeeding self-efficacy in 95 postpartum women from southern Brazil using the BSES-SF scale. The authors report high self-efficacy levels (93.7% of participants) and identify several maternal factors associated with higher scores, including education level, previous breastfeeding experience, and early initiation. The topic is clinically relevant and the use of a validated instrument is appropriate, though several methodological and interpretative concerns limit the manuscript's current contribution.
Major concerns:
The introduction lacks a comprehensive review of existing literature on breastfeeding self-efficacy. What studies have been conducted previously? What gaps exist in current knowledge? The research rationale needs strengthening. Additionally, lines 58-63 contain detailed instrument description that would be better placed in the methods section.
Several methodological issues require clarification. The timing of self-efficacy assessment relative to delivery is not specified - this is crucial given the dynamic nature of early postpartum experiences. Were all participants assessed at the same post-delivery timepoint? The sampling approach is unclear, and no sample size justification is provided.
The statistical analysis section needs expansion. Given the unbalanced group sizes in some comparisons, the authors should describe how they verified statistical assumptions and whether alternative analytical approaches were considered for small groups.
Interpretation issues:
Throughout the discussion, the authors use language suggesting causation where the cross-sectional design only supports associational findings. This represents a significant overinterpretation that undermines the scientific rigor of the work.
Specific Comments
Lines 98-99: How exactly was "adequacy of prenatal care" and "family support" measured? Were specific validated instruments used or custom questions developed? This needs clarification.
Lines 105-114: Please report the psychometric properties (reliability, validity) of the BSES-SF in both the original validation and the Brazilian Portuguese version. What was the Cronbach's alpha in your sample?
Lines 114-116: The definition of adolescence seems out of place here. Is this relevant to your study population or analysis plan?
Lines 117-123: The statistical methods require more detail. How did you test for normality? What specific tests were used for group comparisons? How were assumptions verified, particularly for smaller subgroups?
Lines 125-158: Consider presenting the demographic characteristics in a table format rather than narrative text for better clarity and professional presentation.
Lines 160-163: You mention that some variables mentioned in methods are not reported in results. Either provide these results or explain why they were omitted.
Lines 233-241: The language here is problematic. Phrases like "contributed to increased self-efficacy" and "factors may have contributed" imply causation that your study design cannot establish. Consider rephrasing as "factors associated with" or "variables that showed relationships with."
Lines 268-270: The statement about early initiation being "essential for continuation" goes beyond what your data can support, as you have no follow-up information on breastfeeding duration.
Lines 278-281: The limitations section is inadequate. Missing are key limitations of cross-sectional designs, the inability to infer causation, lack of long-term follow-up, and potential selection bias from single-center recruitment.
Lines 285-287: The conclusions overstate your findings. "Significantly influence" should be "were associated with." The "favorable prognosis" claim requires longitudinal data you don't have.
Additional Issues
Several variables mentioned in the data collection protocol (lines 94-104) don't appear in your results section. Either present these findings or explain their omission.
The manuscript would benefit from a more systematic approach to the literature review in the introduction. Consider organizing existing knowledge around key themes before identifying the specific gap your study addresses.
Recommendations
- Restructure the introduction to include a more thorough literature review and clearer identification of the research gap
- Move detailed instrument descriptions from introduction to methods
- Clarify timing of assessments and sampling methodology
- Provide sample size justification and describe statistical assumption testing
- Revise discussion language throughout to reflect associational rather than causal relationships
- Expand limitations section to include design constraints
- Modify conclusions to align with study design capabilities
- Consider tabular presentation of demographic data
- Address the gap between methods and results regarding unreported variables
The core contribution of this work - examining factors associated with breastfeeding self-efficacy using a validated instrument - has merit. However, the methodological gaps and interpretative overreach currently limit its scientific contribution. With appropriate revisions, particularly regarding the causal language issues and methodological transparency, this could become a solid addition to the breastfeeding literature.
Author Response
Review 2
Major concerns:
The introduction lacks a comprehensive review of existing literature on breastfeeding self-efficacy. What studies have been conducted previously? What gaps exist in current knowledge? The research rationale needs strengthening. Additionally, lines 58-63 contain detailed instrument description that would be better placed in the methods section.
Response: Thank you for your valuable suggestions, which have been duly incorporated into the article.
Several methodological issues require clarification. The timing of self-efficacy assessment relative to delivery is not specified - this is crucial given the dynamic nature of early postpartum experiences. Were all participants assessed at the same post-delivery timepoint? The sampling approach is unclear, and no sample size justification is provided.
Response: Thank you for highlighting these important methodological considerations. The breastfeeding self-efficacy questionnaire was administered within 24 to 48 hours postpartum. All participants were therefore assessed within this early postpartum window. Regarding the sampling approach, the sample was drawn from a well-defined cohort, as detailed in the Methods section. Although no formal sample size calculation was performed prior to data collection, we have acknowledged the limited sample size as a study limitation in the revised manuscript (see Discussion, lines 291-299).
The statistical analysis section needs expansion. Given the unbalanced group sizes in some comparisons, the authors should describe how they verified statistical assumptions and whether alternative analytical approaches were considered for small groups…
Response: Thank you for your insightful comment regarding the statistical analysis. We have expanded the Statistical Analysis section to detail how assumptions of normality and homogeneity of variance were assessed using Shapiro-Wilk and Levene’s test, respectively. In this context, we were allowed for the use of parametric tests such as the t-test and ANOVA for comparison of means. For comparisons of proportions, especially with small or unbalanced group sizes, Fisher’s exact test was applied. These details have been added to the manuscript (see Methods section, page 3, lines 124-134).
Interpretation issues:
Throughout the discussion, the authors use language suggesting causation where the cross-sectional design only supports associational findings. This represents a significant overinterpretation that undermines the scientific rigor of the work.
Response: Thank you for this important observation. We have carefully revised the discussion to ensure that the language reflects associational findings only, consistent with the cross-sectional design of the study. All instances suggesting causation have been corrected accordingly.
Specific Comments:
Lines 98-99: How exactly was "adequacy of prenatal care" and "family support" measured? Were specific validated instruments used or custom questions developed? This needs clarification.
Response: Thank you for your observation. The adequacy of prenatal care was defined according to the recommendation of the Brazilian Ministry of Health, which establishes a minimum of six prenatal visits. Accordingly, adequacy was assessed based on the number of consultations recorded on the pregnancy cards and in medical records. In contrast, the assessment of family support was based on the participants’ subjective perceptions. Women were asked whether they felt emotionally and/or practically supported by their family members, partner, or the maternity care team during pregnancy and the postpartum period. This approach aimed to capture individual experiences of support and care in relation to breastfeeding. A corresponding clarification has been added to the Methods section (Lines 98-99).
Lines 105-114: Please report the psychometric properties (reliability, validity) of the BSES-SF in both the original validation and the Brazilian Portuguese version. What was the Cronbach's alpha in your sample?
Response: Thank you for your valuable comment. The Breastfeeding Self-Efficacy Scale – Short Form (BSES-SF) originally demonstrated excellent psychometric properties, with Cronbach’s alpha values reported between 0.94 and 0.96, indicating high internal consistency, and evidence of validity in multiple populations.
Regarding the Brazilian Portuguese version, previous studies have reported Cronbach’s alpha values around 0.74 to 0.87, confirming good internal consistency and reliability. The intraclass correlation coefficient (ICC) in these studies ranged from 0.69 to 0.78. We have now included these details in the Methods section (Lines 115-118) to clarify the psychometric properties of the instrument used.
Lines 114-116: The definition of adolescence seems out of place here. Is this relevant to your study population or analysis plan?
Response: Thank you for your observation. We agree that the definition of adolescence was not directly relevant to our study population or analysis plan. Since it follows general guidelines and does not impact the study outcomes, we have removed this section to improve clarity and focus.
Lines 117-123: The statistical methods require more detail. How did you test for normality? What specific tests were used for group comparisons? How were assumptions verified, particularly for smaller subgroups?
Response: Thank you for the opportunity to clarify the statistical methods used in this study.
We have now included detailed information in the Methods section (Lines 125–134). The statistical analysis was conducted considering the assumptions underlying each applied test. For categorical variables analyzed by the chi-square test, the independence of observations was ensured, and expected frequencies in each cell were verified. In cases where expected cell counts were less than 5, Fisher’s exact test was employed to maintain the validity of the results. For comparisons of mean scores across instrument domains, assumptions of normality and homogeneity of variances were assessed. Normality was evaluated using the Shapiro-Wilk test complemented by graphical inspection (e.g., Q-Q plots), while homogeneity of variances was tested using Levene’s test. Based on these assessments, either Student’s t-test or ANOVA was applied, according to the number of groups being compared.
Lines 125-158: Consider presenting the demographic characteristics in a table format rather than narrative text for better clarity and professional presentation.
Response: Thank you for your helpful suggestion. To improve clarity and enhance the professional presentation of our manuscript, we have reformatted Table 1 to include a column with the frequency of the demographic characteristics. This change allows for easier interpretation and comparison of the data.
Lines 160-163: You mention that some variables mentioned in methods are not reported in results. Either provide these results or explain why they were omitted.
Response: Thank you for pointing this out. We have carefully reviewed the manuscript to ensure that all variables described in the Methods section are appropriately reported in the Results. Variables that were not directly related to the study objectives were intentionally excluded from the Results to maintain focus and clarity.
Lines 233-241: The language here is problematic. Phrases like "contributed to increased self-efficacy" and "factors may have contributed" imply causation that your study design cannot establish. Consider rephrasing as "factors associated with" or "variables that showed relationships with."
Response: Thank you for your insightful comment regarding the language used in this section. We agree that the terms implying causation are not appropriate given the observational nature of our study.
Accordingly, we have revised the manuscript to replace phrases such as "contributed to increased self-efficacy" and "factors may have contributed" with more accurate expressions like "factors associated with" or "variables that showed relationships with" breastfeeding self-efficacy. These changes have been incorporated to reflect the correlational nature of the findings.
Lines 268-270: The statement about early initiation being "essential for continuation" goes beyond what your data can support, as you have no follow-up information on breastfeeding duration.
Response: Thank you for your comment. We agree that the statement regarding early initiation being “essential for continuation” was not supported by our data, as we did not include follow-up information on breastfeeding duration. We have revised the text to avoid causal implications and to ensure alignment with the scope of our findings. The necessary revisions have been incorporated into the manuscript.
Lines 278-281: The limitations section is inadequate. Missing are key limitations of cross-sectional designs, the inability to infer causation, lack of long-term follow-up, and potential selection bias from single-center recruitment
Response: We have now revised this section (Lines 291–299) to address key limitations, including the inherent constraints of cross-sectional study designs, the inability to infer causality, the lack of long-term follow-up on breastfeeding outcomes, and the potential for selection bias due to recruitment from a single center. These additions strengthen the transparency and critical appraisal of our findings.
Lines 285-287: The conclusions overstate your findings. "Significantly influence" should be "were associated with." The "favorable prognosis" claim requires longitudinal data you don't have.
Response: Thank you for this important observation. We agree that the use of terms such as “significantly influence” and “favorable prognosis” was not appropriate given the cross-sectional nature of our study and the absence of longitudinal data. We have revised the conclusion to reflect the associative—not causal—nature of our findings, and we have removed any reference to prognostic implications.
Round 2
Reviewer 2 Report
Comments and Suggestions for Authors
Thank you for your careful and thoughtful revisions. I appreciate the improvements you have made to the manuscript. All of my previous concerns have been adequately addressed, and I find the current version acceptable for publication. I wish you success in your future research.